# TaiSu: A 166M Large-scale High-Quality Dataset for Chinese Vision-Language Pre-training

Yulong Liu[1,2] , Guibo Zhu[1,4,5] *, Bin Zhu[3] , Qi Song[3] , Guojing Ge[1] , Haoran Chen[1,4] , Guanhui Qiao[1,4] , Ru Peng[2] , Lingxiang Wu[1] , and Jinqiao Wang[1,4,5]

[1]Institute of Automation, Chinese Academy of Sciences (CASIA), Beijing, China
[2]Institute of Artificial Intelligence and Robotics, Xi'an Jiaotong University, Xi'an, China
[3]School of Artificial Intelligence, Beijing Normal University, Beijing, China
[4]School of Artificial Intelligence, University of Chinese Academy of Sciences, Beijing, China
[5]Wuhan AI Research
*{ lylhubxy, nancypt }@stu.xjtu.edu.cn*
*{ gbzhu, jqwang }@nlpr.ia.ac.cn*
*{ bin.zhu, qi.song }@mail.bnu.edu.cn*
*{ guojing.ge, chenhaoran2022,qiaoguanhui2021, lingxiang.wu }@ia.ac.cn*

## Abstract

Vision-Language Pre-training (VLP) has been shown to be an efficient method to improve the performance of models on different vision-and-language downstream tasks. Substantial studies have shown that neural networks may be able to learn some general rules about language and visual concepts from a large-scale weakly labeled image-text dataset. However, most of the public cross-modal datasets that contain more than 100M image-text pairs are in English; there is a lack of available large-scale and high-quality Chinese VLP datasets. In this work, we propose a new framework for automatic dataset acquisition and cleaning with which we construct a new large-scale and high-quality cross-modal dataset named as TaiSu, containing 166 million images and 219 million Chinese captions. Compared with the recently released Wukong dataset, our dataset is achieved with much stricter restrictions on the semantic correlation of image-text pairs. We also propose to combine texts collected from the web with texts generated by a pre-trained image-captioning model. To the best of our knowledge, TaiSu is currently the largest publicly accessible Chinese cross-modal dataset. Furthermore, we test our dataset on several vision-language downstream tasks. TaiSu outperforms BriVL by a large margin on the zero-shot image-text retrieval task and zero-shot image classification task. TaiSu also shows better performance than Wukong on the image-retrieval task without using image augmentation for training. Results demonstrate that TaiSu can serve as a promising VLP dataset, both for understanding and generative tasks. More information can be referred to `https://github.com/ksOAn6g5/TaiSu`.

## 1 Introduction

Vision-Language Pre-training (VLP) is a new paradigm to obtain the foundation models with large-scale parameters[1]. It has achieved great success on different vision-language downstream tasks such as Image-Text Retrieval, Image-Captioning, and Visual Question Answering. Its success is not only attributed to the appearance of advanced models like BERT[2] and ViT[3], but also to the use of large-scale multi-modal datasets. For example, CLIP[4] was trained with 400M image-text pairs,

---

*Corresponding author.

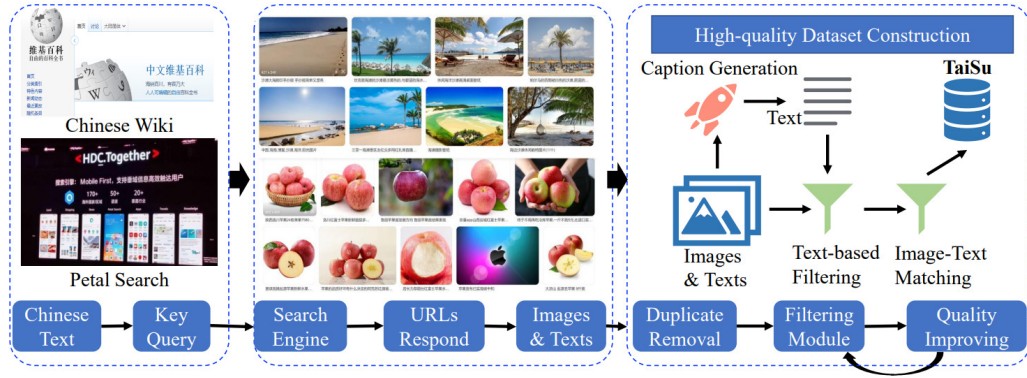

Figure 1: An overview of our framework. This framework filters the collected data with text-based filtering module, image-text-retrieval-based filtering module, and Image-Captioning-based module

SimVLM[5] with 1.8B image-text pairs, and M6[6] with more than 60.5M image-text pairs. Table 1 shows an overview of the popular datasets in the VLP domain. We can see that most of the large-scale multi-modal datasets are based on English corpus. In addition, there exists a huge difference between Chinese and English in terms of culture. Therefore, the construction of a large-scale Chinese multi-modal dataset is necessary. Currently, Chinese multi-modal datasets that cover more than 50M image-text pairs only include M6-corpus[6], Wukong[7] and WudaoMM[8]. However, M6-corpus has not been publicly released and WudaoMM only released 5M image-text pairs. Wukong is currently the largest available Chinese multi-modal dataset, containing 100M image-text pairs. Thus, compared with the English VLP community, there is a lack of large-scale and high-quality public Chinese multi-modal datasets for VLP tasks.

There are several reasons for this problem. First, constructing a large-scale multi-modal dataset of high quality requires relatively high labor and hardware costs. Second, current large-scale Chinese multi-modal datasets are only filtered with the information from a single modality while the mismatched data pairs may hinder the learning process of VLP models. Third, we find that almost all the large-scale multi-modal datasets provide only one description for each image because it's actually pretty hard to get multiple descriptions from the Web for a given image. However, a single description from the Web is usually hard to accurately depict the characterization of the corresponding image.

To facilitate the research of Chinese Vision-language pre-training, we design a new framework for the construction of a large-scale multi-modal dataset. We show the framework in Figure 1. Firstly, in order to guarantee diversity and generalization, we use the 900K most frequent search items from Petal Search Engine and Chinese Wiki as the queries to crawl image-text pairs from the Web. Secondly, after the image-text pairs are collected, the text-based filtering module, the image-text-retrieval-based filtering module, and the Image-Captioning-based text augmentation module are applied to the data. Especially, The image-text-retrieval-based filtering module is a self-supervised image-text matching process. To be more specific, we pre-train a model for the image-text retrieval task on the crawled image-text pairs and then filter the very same dataset using the pre-trained model. The function of the text augmentation module is to generate auxiliary captions for the crawled images and combine these two kinds of texts for vision-language pre-training. Our experiments show that this filtering framework can improve the zero-shot performance of the pre-trained models.

With the proposed framework, we construct a large-scale and high-quality Chinese multi-modal dataset denoted as TaiSu, containing about 166M images and 219M Chinese captions. Our main contributions are summarized as follows:

(1) We propose an effective and automatic dataset acquisition and cleaning framework which includes the text-based filtering module, image-text-retrieval-based filtering module, and Image-Captioning-based text augmentation module. The framework can continuously improve the quality of the dataset in a self-supervised manner.

(2) With the help of the automatic filtering methods, we construct a large-scale high-quality cross-modal dataset, named as TaiSu, which contains about 166M images and 219M captions. To

Table 1: An overview of datasets for VLP model pre-training. Our proposed TaiSu is the largest available dataset for Chinese vision-language pre-training.

| Dataset | Language | Available | Size |
|---|---|---|---|
| CC3M[9] | English | Yes | 3,000,000 |
| M5Product[10] | English | Yes | 5,000,000 |
| JFT-300M[11] | English | No | 300,000,000 |
| WIT[12] | multilingual | Yes | 11,500,000 |
| CC12M[13] | English | Yes | 12,000,000 |
| YFCC100M[14] | English | Yes | 99,200,000 |
| LAION-400M[15] | English | Yes | 400,000,000 |
| JFT-3B[16] | English | No | 3,000,000,000 |
| IG-3.5B-17K[17] | English | No | 3,500,000,000 |
| LAION-5B[18] | multilungual | Yes | 5,850,000,000 |
| Product1M[19] | Chinese | Yes | 1,000,000 |
| WudaoMM[8] | Chinese | Yes | 5,000,000 |
| M6-Corpus[6] | Chinese | No | 60,500,000 |
| WuKong[7] | Chinese | Yes | 101,483,885 |
| **TaiSu(Ours)** | Chinese | Yes | **166,000,000** |

the best of our knowledge, TaiSu is currently the largest public cross-modal dataset based on Chinese corpus. To help the development of the image retrieval community, we will also release the pre-extracted image embeddings to facilitate future research.

(3) Our extensive experiments demonstrate the effectiveness of the proposed framework. Meanwhile, it brings a new insight that combining the raw Web image-text pairs with the generated image-text pairs can greatly improve the zero-shot performance on the image-text retrieval task.

.

## 2 Related work

**Vision-Language pre-training datasets**    In the early stage of VLP, multimodal datasets are usually manually annotated for specific downstream tasks. For example, MSCOCO[20] and Flickr30K[21] are constructed for the image-captioning task and provide multiple descriptions for one image, VQA[22] annotates images in a question-and-answer format for the VQA task. These datasets require a lot of manpower to complete, which results in a relatively small scale and a lack of diversity. Then some works try to create datasets programmatically where the image-text pairs are automatically crawled from the Web followed by filtering processes. CC3M[9] is constructed using a Flume[23] framework and contains 3M image-text pairs, then CC12M[13] scales up VLP data to 12 million by relaxing overly strict filters in CC3M. With the development of hardware and distributed training frameworks, the VLP datasets have attended a scale of several hundred million or even several billion [14, 15, 11, 16, 6, 18]. In [15], the authors proposed LAION-400M, which contains 400 million English image-text pairs. Then, in [18], the same research group scaled up the data to 5 billion and released LAION-5B. Before TaiSu, Wukong[7] was the largest Chinese multimodal dataset, containing 100M image-text pairs. By comparison, the number and size of accessible Chinese multimodal datasets are inadequate. In this paper, we will release TaiSu to bridge this gap.

**Vision-Language pre-training Models**    There exist two typical VLP model architectures, i.e., single-stream and dual-stream. Single-stream models[24, 25, 26, 27, 28] usually concatenate an image and a text as a single data stream and use a transformer[29] to model the interactions between the image and the text. This kind of architecture often employs language modeling, masked language modeling, and image-text matching to learn the joint representation of image and text. In contrast, dual-stream models like CLIP[4], FILIP[30], and ALIGN[31] use two encoders to process images and texts respectively. Dual-stream models are usually trained with the contrastive loss[32] to align the semantics of the paired image and text in the same embedding space. To this end, there exist two approaches to computing the similarity between an image and a text, i.e., global similarity[4] and token-wise similarity[30]. In our experiments, we adopt dual-stream architecture and global similarity due to simplicity and efficiency.

# 3 TaiSu dataset creation

TaiSu is a combination of the image-text pairs collected from the Web and the texts generated by an image-captioning model. The collection and processing steps are described in detail in the following subsections.

## 3.1 Data collection

In order to create a dataset rich in visual concepts and vocabulary, a query list is constructed from a list of query terms provided by Huawei Petal Search Engine and Chinese Wiki. We filter the queries according to their frequencies and get rid of those meaningless words, which finally results in a set of 900K queries. Then, we submit the queries to Baidu Image Search Engine and Bing Search Engine to get image URLs and corresponding caption information. And we set the maximum number of samples that each query can get to 2000. After that, images are downloaded with these image URLs. In total, 177M raw image-text pairs are collected.

## 3.2 Text-based filtering

We allow text between 2 and 50 characters in our dataset. And we construct a list of sensitive words; texts with those words are discarded. As many collected sentences contain information about the source that has no semantic connection with the corresponding image, like "网易"(Netease), "新浪博客"(Sina blog), "京东商城"(JD Mall), we delete these words from the raw texts. To protect the privacy of the individuals appearing in the text, we perform person-name substitutions by substituting person names with a special token "⟨人名⟩" (⟨Person name⟩).

## 3.3 Boosting with self-supervised cross-modal matching

Text-based filtering can't assure that each sentence is correlated with corresponding image. To boost the correlation on the dataset level, we propose to filter image-text pairs with a self-supervised cross-modal matching process. It means that we train a model for image-text retrieval task using all the collected data and then use this model to filter out those image-text pairs with low correlation. To be more specific, we train a CLIP-like dual-stream model that can encode images and Chinese sentences into the same embedding space. Then embeddings of the images and texts are computed with this model. Unlike LAION-400M[15] where a fixed threshold of cosine similarity is used to filter out illegal contents, we treat the filtering process as a retrieval task. We calculate the cosine similarity between images and texts in the same batch. As the batch size is usually pretty large and varies from machine to machine, we propose to retrieve texts and images only in the range of a smaller window. If the caption in an image-text pair best matches that image or vice versa, we take this image-text pair as qualified data, otherwise this image-text pair is discarded. In this way, we can roughly control the filtering quality by setting different window size, i.e., the larger the window size is, the higher the average matching degree of the final dataset is. For TaiSu, we set the window size to 120. Finally, we get a total of 133M Web image-text pairs.

## 3.4 Boosting with Image-Captioning

To the best of our knowledge, TaiSu is the first large-scale Chinese multi-modal dataset that combines the texts from the Web and the texts generated by machine. Although there are several Chinese cross-modal datasets such as COCO-CN[33] and AIC-ICC[34] that give multiple captions for one image, their scales of data are usually too small for pre-training. Furthermore, it's hard to directly get multiple descriptions from the Web for each image, so we adopt an image caption generator to generate a supplementary description for the collected images. Although the texts generated by machine are usually bland, they are on average highly correlated with the input images. In order to get captions with high accuracy and diversity, We use the OFA-large[35] model which is pre-trained with large-scale English image-text pairs and obtains a CIDEr score of 149.6 on COCO Captions[20]. Then the generated English captions are translated into Chinese with machine translation. Since these English sentences are usually pretty simple, the translation quality can be basically assured. We delete those generated captions paired with more than two thousand images because they may be too generic or irrelevant to the content of the images. To further improve the data quality, we also apply the aforementioned image-text-retrieval-based filtering process to the generated texts.

Table 2: Statistics of TaiSu's text data.

| Text source | Text | Unique Text | Noun Phrase |
|---|---|---|---|
| Web Data | 133150708 | 96146658 | 5032017 |
| Generated Data | 85957428 | 18644050 | 101036 |
| All Data | 219108136 | 114786961 | 5062557 |

## 3.5 Characteristics of TaiSu

**Basic statistics** After the processing stages mentioned above, we get a dataset containing 166M images,133M texts from Webpages, and 86M texts generated by the Image-Captioning mechanism. And we also take the diversity of sentence patterns and the richness of visual concepts into consideration, we count the number of different sentences and use the Chinese text segmentation module jieba[2] to generate words and count the number of nouns. The statistics are shown in Table 2. There are 115M different sentences and more than 5M different nouns in the Web texts, while the generated texts have much lower diversity.

**Human evaluation** We compute a rough estimate of the precision on two hundred image-text pairs randomly sampled from TaiSu. Specifically, we ask two annotators to rate how well the given caption fits the image on a 1–4 scale: 1(no fit),2(barely fit),3(good fit, but with poor fluency),4(perfect). It turns out that 62.57% of the captions get a score of no less than 3, and only 11.17% of the captions are estimated as no fit. More details about TaiSu can be found in Appendix.

## 4 Methodology

**Large-scale vision-language contrastive learning** Contrastive learning has been shown to be an effective method for vision and language understanding[4, 36]. To conduct vision-language contrastive learning, we adopt a dual-stream architecture where two encoders are respectively used to embed images and texts into the same embedding space. The model is forced to make the embeddings of paired samples close and make the embeddings of unpaired samples far apart. Given a batch of $N$ image-text pairs, there are $N^2$ possible image-text pairs, but only $N$ pairs are regarded as positive samples and the rest $N^2 - N$ pairs are regarded as negative. Formally, each image in $\{x_k^I\}_{k=1}^N$ is encoded by the visual encoder and normalized to get its embedding vector $z_k^I \epsilon R^D$ , and each text in $\{x_k^T\}_{k=1}^N$ is encoded by the textual encoder and normalized to get its embedding vector $z_k^T \epsilon R^D$, where $x_k^I(resp.x_k^T)$ means the image (resp. text) of the $k$-th pair in the same batch, and $N$ is the number of samples in the batch. Therefore, the embeddings of images and texts are distributed on a unit hypersphere. Then we compute the image-to-text and text-to-image similarities:

$$s_{i,j}^{i2t} = z_i^I * z_j^T, s_{i,j}^{t2i} = z_i^T * z_j^I,$$

where *is the dot product, $s_{i,j}^{i2t}$ is the image-to-text similarity between the image of the $i$-th pair and the text of the $j$-th pair, $s_{i,j}^{t2i}$ represents the text-to-image similarity between the text of the $i$-th pair and the image of the $j$-th pair. Contrastive losses of this batch are calculated as:

$$p_i^{i2t} = \frac{exp(s_{i,i}^{i2t}/\sigma)}{\sum_{j=1}^N exp(s_{i,j}^{i2t}/\sigma)}, p_i^{t2i} = \frac{exp(s_{i,i}^{t2i}/\sigma)}{\sum_{j=1}^N exp(s_{i,j}^{t2i}/\sigma)},$$

$$L^{i2t} = -\frac{1}{N}\sum_{i=1}^N logp_i^{i2t}, L^{t2i} = -\frac{1}{N}\sum_{i=1}^N logp_i^{t2i},$$

where $L^{i2t}$ represents the image-to-text loss and $L^{t2i}$ is the text-to-image loss, $\sigma$ is a learnable temperature parameter. The total loss $L$ of this training batch is then computed as:

$$L = \frac{1}{2}(L^{i2t} + L^{t2i}).$$

---

[2]`https://github.com/fxsjy/jieba`

**Model architectures**    As mentioned above, we use a dual-stream architecture that consists of a visual encoder and a textual encoder. For the visual encoder, we utilize either a RN101[37] model or a ViT-B/32[3] model as the backbone. We follow the same setting of CLIP[4] for these two kinds of variants. To get the global feature embedding of an image, an attention-pool layer is added after the ResNet blocks of the RN101 variant; and for the ViT-B/32 variant, the output corresponding to the [CLS] token is linearly projected to the common embedding space. The global feature embeddings of images are normalized with L2-normalization. For the textual encoder, we use a 12-layer standard decode-only transformer with 8 attention heads and a hidden state dimension of 512. We use SentencePiece[38] for the tokenization of Chinese sentences, and the pre-trained tokenizer with a vocabulary size of 50000 is provided by CogView[39]. We add two special tokens (i.e., [CLS] and [SEP]) at the beginning and ending of each Chinese sequence. The output corresponding to the [SEP] token is used as the global feature of the given text and is linearly projected into the embedding space and normalized with L2 normalization.

**Locked-image Text Tuning**    In order to speed up the training process and save computing resources, we take advantage of the Locked-image Text Tuning technique (LiT-tuning)[40] with which we only need to train the textual encoder and keep the pre-trained visual encoder fixed. And the experiments of Wukong[7] have shown that a visual encoder trained with English corpus can transfer well to Chinese corpus.

## 5    Experiments

### 5.1    Implementation Details

We use the RN101 and ViT-B/32 pre-trained for CLIP[4] as our locked image encoders. To speed up the training process, the global image feature embeddings are extracted before training which means that we don't employ any image data augmentation as Wukong does. All of our LiT-tuning models are trained with 200 Sugon DCUs for 120 epochs and the batch size on each DCU is set to 860. We update the parameters using AdamW [41] with $\beta1 = 0.9$, $\beta2 = 0.999$, $\epsilon = 10^-8$ , and weight decay multiplier 0.2. And we use the cosine learning rate schedule with a linear warmup[42].

To verify the effects of the filtering based on image-text matching and the effects of the use of the generated texts, we use three types of data which are denoted as follows: RD(the raw Web data containing 177M image-text pairs), $TS_{Web}$ (the Web data in TaiSu,containing 133M Web image-text pairs) and $TS_{all}$ (the combination of $TS_{Web}$ and 86M generated image-text pairs). When we combine the Web texts and generated texts to train models, we randomly choose either the Web text or the generated text for an image in every epoch.

### 5.2    Zero-shot Image-text Retrieval

There exist two subtasks in image-text retrieval task: image-to-text retrieval and text-to-image retrieval. In both of these two subtasks, the model is expected to find the target sample from another modality in a pool of candidates given either an image or a text as query. We evaluate the zero-shot performance of the models pre-trained with TaiSu on 4 datasets, including Flickr8K-CN[43], Flickr30K-CN[21], COCO-CN[33], MUGE[3]. We report recall of top K candidates (Recall@K) with K = 1, 5, 10 for the two subtasks on all the datasets, except for MUGE, which only has the text-to-image setting. we use the average of Recall@K, i.e., Mean Recall (MR), for the final comparison. The results for Flickr8K-CN, Flickr30K-CN and COCO-CN are computed on the test sets,and the results for MUGE are computed on the validation set.

In Table 3, we compare our pre-trained models with BriVL[44] and Wukong[7] on the zero-shot image-text retrieval task. Both the two kinds of models are trained on large-scale Chinese cross-modal datasets. We can see that TaiSu significantly outperforms BriVL on all the test datasets. And in comparison with Wukong, TaiSu gets better performances on nearly all the datasets, except for MUGE. Note that TaiSu gets such results without applying image augmentation as Wukong does. These results can show the effectiveness of TaiSu. Further more, We attribute Taisu's success on image-text retrieval task to the combination of Web texts and generated texts, as is shown in Table 5. On the one hand, the addition of generated texts will increase the number of image captions that can

---

[3]`https://tianchi.aliyun.com/muge`

Table 3: Results of zero-shot image-text retrieval on kinds of datasets. Our models shown in this table are trained on TS$_{\text{all}}$. The results of Wukong are taken from [7], and those of BriVL are obtained with its pretrained model weight and implementation code.

| Dataset | Method | Image-to-Text | | | Text-to-Image | | | MR |
| --- | --- | --- | --- | --- | --- | --- | --- | --- |
| | | R@1 | R@5 | R@10 | R@1 | R@5 | R@10 | |
| Flickr8K-CN | BriVL[44] | 13.4 | 31.2 | 40.7 | 8.0 | 20.7 | 29.5 | 23.9 |
| | Wukong$_{\text{ViT-B}}$[7] | 55.4 | 82.3 | 90.0 | 43.2 | 71.3 | 81.3 | 70.6 |
| | Wukong$_{\text{Swin-L}}$[7] | 47.2 | 78.8 | 87.6 | 36.6 | 64.8 | 76.2 | 65.2 |
| | Ours$_{\text{RN101}}$ | 55.1 | 82.6 | **90.9** | 44.9 | 74.2 | **84.3** | 72.0 |
| | Ours$_{\text{ViT-B}}$ | **57.6** | **83.4** | 90.6 | **45.4** | **74.4** | 84.1 | **72.6** |
| Flickr30K-CN | BriVL[44] | 17.7 | 42.3 | 54.3 | 10.3 | 27.5 | 37.9 | 31.7 |
| | Wukong$_{\text{ViT-B}}$[7] | **66.2** | 88.7 | 94.3 | 45.7 | 73.8 | 82.2 | 75.1 |
| | Wukong$_{\text{Swin-L}}$[7] | 58.7 | 86.7 | 92.7 | 40.9 | 68.0 | 78.4 | 70.9 |
| | Ours$_{\text{RN101}}$ | 65.3 | 88.6 | 94.1 | **51.2** | **79.1** | **89.5** | 77.6 |
| | Ours$_{\text{ViT-B}}$ | 65.6 | **90.1** | **94.9** | 49.9 | 78.9 | 87.0 | **77.7** |
| COCO-CN | BriVL[44] | 17.1 | 41.7 | 57.5 | 14.8 | 39.0 | 54.2 | 37.4 |
| | Wukong$_{\text{ViT-B}}$[7] | 48.3 | 77.8 | 88.8 | 49.2 | 79.4 | 87.9 | 71.9 |
| | Wukong$_{\text{Swin-L}}$[7] | 47.3 | 78.0 | 88.3 | 46.4 | 77.0 | 87.6 | 70.8 |
| | Ours$_{\text{RN101}}$ | **54.1** | **82.8** | **91.8** | 54.3 | 82.7 | **92.4** | **76.4** |
| | Ours$_{\text{ViT-B}}$ | 52.5 | 81.5 | 91.4 | 53.6 | **83.7** | 92.4 | 75.9 |
| MUGE | BriVL[44] | - | - | - | 12.7 | 30.9 | 41.8 | 28.5 |
| | Wukong$_{\text{ViT-B}}$[7] | - | - | - | 33.4 | 59.3 | 69.7 | 54.1 |
| | Wukong$_{\text{Swin-L}}$[7] | - | - | - | **34.5** | **60.6** | **71.2** | **55.5** |
| | Ours$_{\text{RN101}}$ | - | - | - | 27.5 | 53.9 | 64.8 | 48.7 |
| | Ours$_{\text{ViT-B}}$ | - | - | - | 29.7 | 57.0 | 67.4 | 51.4 |

be used for training; On the other hand, for those images that have both Web text and generated text, the generated text can provide auxiliary information for the model to learn a better representation.

## 5.3 Zero-shot Image Classification

To evaluate TaiSu on zero-shot image classification task, We report the performances of our pre-trained models on 8 datasets[45, 45, 46, 47, 48, 49, 50, 51]. All the class labels are translated from English into Chinese by online machine translation and then are manually refined. Following the common practices, we design some text prompts for each dataset to provide the labels with some contextual information[4]. And we use the mean textual representation of different prompt templates as the global textual representation for each class label. We report the top1 Recall for all our pre-trained models.

As shown in Table 4, the models pretrained on TaiSu outperform BriVL by a large margin on almost all the datasets used for testing, which implies that TaiSu covers a wide range of domains and has fairly good generalization ability.

## 5.4 Text-to-image Generation

As a direct way to show what the model can learn from our dataset, We use VQGAN[52] to generate different types of images under the guidance of the dual-stream model pre-trained on TaiSu. To be more specific, We first randomly sample a sequence of tokens from the codebook of VQGAN, and the decoder of VQGAN will generate an image with the sampled tokens. Next, we compute the embedding of the generated image and that of the given text prompt with our pretrained ViT-B/32 variant. We optimize the image by minimizing the squared spherical distance between the embedding of the candidate image and the embedding of the text prompt. The resultant gradients are back-propagated to update the image token sequence. All the model parameters are frozen during the generation process.

We first try to generate images of different artistic styles. In Figure 2, we show some generated samples. We can see that the model can accurately understand different painting styles and different

Table 4: Zero-shot Image classification results on different datasets. RN101 and ViT-B variants are trained on different data. RD means the raw Web data; $TS_{Web}$ means the Web data filtered by image-text retrieval; $TS_{all}$ means the filtered Web data plus data generated by Image-Captioning

| Dataset | Metric | BriVL[44] | RN101 | | | ViT-B | | |
|---|---|---|---|---|---|---|---|---|
| | | | RD | $TS_{Web}$ | $TS_{all}$ | RD | $TS_{Web}$ | $TS_{all}$ |
| CIFAR10 | R@1 | 72.3 | 79.8 | 81.0 | 74.1 | 87.9 | **89.2** | 85.5 |
| CIFAR100 | R@1 | 35.9 | 43.3 | 43.0 | 43.0 | 55.5 | **56.7** | 55.4 |
| Caltech101 | R@1 | 72.0 | 71.4 | 71.8 | 73.5 | 72.6 | 73.2 | **74.2** |
| Caltech256 | R@1 | 58.0 | 68.5 | 69.9 | 70.2 | 69.1 | 71.0 | **72.0** |
| DTD | R@1 | 18.8 | 24.9 | 26.2 | **29.8** | 26.5 | 29.3 | 28.7 |
| Flowers | R@1 | 18.4 | 24.3 | 23.3 | 21.3 | 23.8 | **30.6** | 22.4 |
| EuroSAT | R@1 | 25.5 | 29.9 | 23.6 | 39.7 | 36.0 | 29.5 | **52.6** |
| ImageNet | R@1 | 24.3 | 33.3 | 34.2 | 33.4 | 33.7 | **35.4** | 34.0 |
| AVG | | 40.65 | 46.93 | 46.63 | 48.13 | 50.64 | 51.86 | **53.10** |

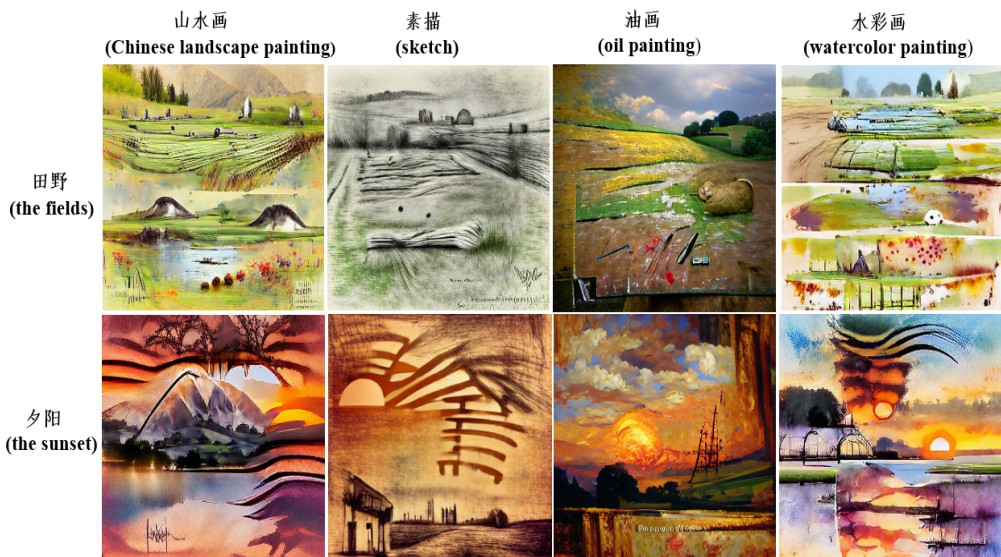

Figure 2: Generated samples of different artistic styles. The model pretrained on TaiSu can correctly guide VQGAN to generate different types of images

scenes. We also find that our model is able to correctly understand and draw some very abstract conceptions like "重复的几何图案" ("repeated geometric patterns"), which implies that TaiSu dataset is rich in visual conceptions and can be applied in different fields. More generated samples can be found in Appendix.

## 5.5  Ablation Study

**Influence of the filtering based on image-text Retrieval**    To verify the effects of the filtering based on image-text retrieval, we respectively train models with the raw Web data pairs (RD) and the filtered Web data pairs($TS_{Web}$). The results are shown in Table 4 and Table 5. It turns out that such a filtering method can improve the zero-shot performance both on the classification task and image-text retrieval task. Especially, on the image-text retrieval task, all metrics are improved after filtering. This can be explained by the removal of mismatched data pairs. Meanwhile, we also observe a significant drop in the top1 recall for the classification task on EuroSAT. This phenomenon may be caused by the deterioration of unbalanced distribution for those categories with few samples.

Table 5: Zero-shot image-text retrieval results on different datasets. The models are trained on different data. RD means the raw Web data; $TS_{Web}$ means the Web data that is filtered by image-text retrieval; $TS_{all}$ means the filtered Web data with data generated by Image-Captioning.

| Dataset | Model | Image-to-Text | | | Text-to-Image | | | MR |
| | | R@1 | R@5 | R@10 | R@1 | R@5 | R@10 | |
|---|---|---|---|---|---|---|---|---|
| Flickr8k | RD-RN101 | 41.0 | 69.0 | 80.0 | 28.3 | 54.4 | 66.8 | 56.6 |
| | $TS_{Web}$-RN101 | 44.2 | 69.2 | 80.2 | 29.4 | 56.0 | 68.3 | 57.9 |
| | $TS_{all}$-RN101 | 55.1 | 82.6 | **90.9** | 44.9 | 74.2 | **84.3** | 72.0 |
| | RD-ViT | 35.5 | 63.3 | 76.8 | 26.9 | 52.4 | 64.7 | 53.8 |
| | $TS_{Web}$-ViT | 40.8 | 70.1 | 80.6 | 31.3 | 57.4 | 69.0 | 58.2 |
| | $TS_{all}$-ViT | **57.6** | **83.4** | 90.6 | **45.4** | **74.4** | 84.1 | **72.6** |
| Flickr30k | RD-RN101 | 47.0 | 75.5 | 85.3 | 31.9 | 61.5 | 72.9 | 62.4 |
| | $TS_{Web}$-RN101 | 48.5 | 77.4 | 86.2 | 35.2 | 63.5 | 73.8 | 64.1 |
| | $TS_{all}$-RN101 | 65.3 | 88.6 | 94.1 | **51.2** | **79.1** | **89.5** | 77.6 |
| | RD-ViT | 42.3 | 71.6 | 82.8 | 29.7 | 57.1 | 68.6 | 58.7 |
| | $TS_{Web}$-ViT | 49.8 | 77.3 | 87.2 | 34.7 | 63.0 | 73.0 | 64.2 |
| | $TS_{all}$-ViT | **65.6** | **90.1** | **94.9** | 49.9 | 78.9 | 87.0 | **77.7** |
| MUGE | RD-RN101 | - | - | - | 26.7 | 52.4 | 63.8 | 47.6 |
| | $TS_{Web}$-RN101 | - | - | - | **30.1** | 56.4 | **67.4** | 51.3 |
| | $TS_{all}$-RN101 | - | - | - | 27.5 | 53.9 | 64.8 | 48.7 |
| | RD-ViT | - | - | - | 27.6 | 53.6 | 65.1 | 48.8 |
| | $TS_{Web}$-ViT | - | - | - | 29.7 | **57.0** | **67.4** | **51.4** |
| | $TS_{all}$-ViT | - | - | - | 28.5 | 54.2 | 65.2 | 49.3 |
| COCO-CN | RD-RN101 | 41.8 | 71.6 | 84.2 | 39.8 | 69.9 | 81.5 | 64.8 |
| | $TS_{Web}$-RN101 | 43.4 | 75.1 | 85.7 | 41.8 | 70.6 | 84.8 | 66.9 |
| | $TS_{all}$-RN101 | **54.1** | **82.8** | **91.8** | **54.3** | 82.7 | **92.4** | **76.4** |
| | RD-ViT | 40.8 | 70.3 | 83.3 | 37.2 | 68.6 | 82.2 | 63.7 |
| | $TS_{Web}$-ViT | 44.4 | 73.3 | 84.8 | 39.8 | 72.3 | 83.9 | 66.4 |
| | $TS_{all}$-ViT | 52.5 | 81.5 | 91.4 | 53.6 | **83.7** | **92.4** | 75.9 |

**Influence of the addition of generated texts**  We compare the performances of models trained with $TS_{Web}$(filtered Web data) and $TS_{all}$(filtered Web data and generated data) in Table 4 and Table 5. The models trained with $TS_{all}$ have much better zero-shot image-text retrieval performance, and the zero-shot classification performance on EuroSAT is boosted from 29.5 to 52.6 for the ViT model. Therefore, the addition of generated texts can indeed bring some benefits to downstream tasks. However, on some test datasets such as CIFAR10 and CIFAR100, the zero-shot classification performance shows a slight drop. This may be related to the limitation of the Image-captioning model used in our experiments. The Image-captioning model, i.e, OFA[35], is trained with about 15M image-text pairs and has a limited domain. So it tends to generate some generic captions. A better way is to pre-train a generative model on our large-scale collected image-text pairs, and then use this model to generate captions for the construction of the dataset. This will be left for future work.

# 6   Conclusion

In this work, we propose a new framework for the automatic cleaning of large-scale cross-modal datasets and construct the largest publicly available Chinese cross-modal dataset, TaiSu. Our experiments demonstrate that it's suboptimal to directly train VLP models on raw Web data. Filtering the Web data with semantic correlation constraints can boost the performance of pre-trained models to some extent. We also find that it's helpful to combine generated texts with Web texts for image-text retrieval tasks. Finally, the results show that TaiSu has the potential to benefit different multimodal downstream tasks. In the future, we will further improve the effectiveness of TaiSu and increase the data scale continuously.

## Acknowledgments and Disclosure of Funding

This work was supported by National Key R&D Program of China under Grant No. 2021YFE0205700, National Natural Science Foundation of China (No.62076235,62002356, 61976210, 62002357, 62176254, 61876086, 62006230, 2022M713363), sponsored by Zhejiang Lab (No.2021KH0AB07). The numerical calculations in this study were carried out on the ORISE Supercomputer.

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
