# OpenReview forum: "TaiSu: A 166M Large-scale High-Quality Dataset for Chinese Vision-Language Pre-training"
_NeurIPS.cc/2022/Track/Datasets_and_Benchmarks — NeurIPS 2022 Datasets and Benchmarks _

### Official Review · Reviewer_VN2W · 2022-07-18

**Rating:** 6
**Confidence:** 3
**Correctness:** Yes. The dataset collection process s…
**Clarity:** Overall, I feel it is easy to follow …

**Strengths:**

1. The dataset will be useful towards better evaluating cross-modal image-text tasks, especially for the ones who based on Chinese corpus.

2. The dataset is quite large and diverse in comparison with existing Chinese cross-modal datasets, such as Wudao, M6, and WuKong. The comparison with existing datasets is clear.

3. The authors have done a good job implementing several classes of baselines and the results look interesting.

**Weaknesses:**

1. There could be more discussion regarding the annotation process since there may be possible biases resulting from the annotation process given that different annotators follow different distributions. More discussion regarding checks for the accuracy of annotations would also help the paper.

2. There are some typos in this paper as follows:
    - L63 cross-model → cross-modal
    - L67 cross-model → cross-modal
    - L80 result → results

**Additional Feedback:**

none.

**Documentation:**

Yes. Though there could be more discussion regarding the annotation process. More discussion regarding checks for the accuracy of annotations would also help the paper.

**Ethics:**

No.

**Relation To Prior Work:**

Yes. The comparison with existing datasets is clear.

**Summary And Contributions:**

This paper proposes the largest available Chinese Vision-Language Pre-training dataset -- TaiSu. The contributions of this paper can be summarized threefold:

1. This paper proposes a new framework for automatic dataset cleaning, which includes text based filtering, cross-model semantic based filtering, and image captioning. The proposed framework can continuously improve the quality of the dataset in a self-supervised manner.

2. The authors construct the currently largest (166M) public Chinese cross-modal image-text dataset, TaiSu. They also release the pre-extracted image feature embeddings using RN101 and ViT to facilitate future researchers in the image retrieval community.

3. The experimental results demonstrate the effective of the proposed framework, and bringing improvements for zero-shot image-text retrieval by combining the raw Web and the generated image-text pairs.

---

> ### Author Response · Authors · 2022-08-23
> **Rough estimate of the precision has been conducted.**
>
> Thanks for your review comments!
> The Image-captioning model can bring biases, that’s true. We have also mentioned this problem in the paper.  OFA, the image captioning model used in our experiments, is currently a state-of-the-art model for this task and was trained on more than 15M data. In the dataset creation section, we have computed a rough estimate of the precision of TaiSu’s data.  To compute the accurate precision for such a huge dataset is a difficult thing. It’s also mentioned in the paper that a better way is to pre-train an image-captioning model with our large-scale dataset and use it to annotate our data. But that will be left for future work.   Thanks for your advice, And thanks for pointing out the spelling mistakes.

---

### Official Review · Reviewer_HiVb · 2022-07-21
**A High-Quality Dataset for Chinese V-L Pretraining**

**Rating:** 7
**Confidence:** 4
**Clarity:** Yes.

**Strengths:**

1, V-L Pretraining is a hot topic and the community is lack of Chinese benchmarking.
2. Another Chinese large-scale benchmarking dataset using more restricted constraints.
3. Generated some samples of different artistic styles.

**Weaknesses:**

1. Need more comparisons with more VL-pretraining methods other than just CLIP. Methods in Table 4 and Table 5 are all CLIP pretraining methods with different backbones and different dataset constructions. In order to have a better benchmarking, the Authors may compare with more VL-pretraining methods such as FILIP, DeCLIP, DeFILIP, and so on.
2. The generation task is very interesting. Can you provide a more quantitive analysis on generation tasks with some comparing methods such as DallE and GLIDE? For example, provide evaluation metrics like  FID, IS, and Clip-score on MS-COCO CN?
3. By the way, what is Mental Search in Fig.1?

**Additional Feedback:**

None.

**Correctness:**

The performance of Wukong in Table 3 seems to be out of date. For example, the Flickr30K-CN of Wukong is mean recall(MR) 93.4 other than the reported 75.1. The ImageNet zero-shot top-1 acc performance for Wukong is 58% (not 35%) You may update the numbers from wukong latest arxiv version.

**Documentation:**

The license of the dataset is missing. The authors need to provide them in the appendix.

**Ethics:**

None.

**Relation To Prior Work:**

Yes. The comparison with Wukong may need to be updated.

**Summary And Contributions:**

1. Construct a new large-scale and high-quality image-text pair dataset named as TaiSu, containing 166 million images with captions.
2. Add more restricted constraints on filtering the dataset.
3, Add some model-generated captions in the dataset and show some improvements.

---

> ### Author Response · Authors · 2022-08-23
> **The figures of Wukong are explained.**
>
> Thanks for your time spent on the review process.
>
> 1), In this work, we try to explore the methods to construct a large-scale cross-modal dataset of high quality. So we pretrain our models with different data to verify the effectiveness of the proposed methods. As you know, pretraining on such a huge dataset is expensive and time-consuming, and providing a comprehensive Benchmarking for different training methods is not our main goal in this paper, We will adopt your advice and implement more experiments in our future work.
>
> 2), In this work, we have shown some images generated by VQ-GAN under the guidance of our pretrained dual-stream model. We regard it as a method to visualize the knowledge learned from our proposed dataset.  And we think that comparing this method with DALL-E or Cogview is unfair. Because our dual-stream models are trained to understand the high level feature(semantic) of images while DALL-E and CogView are directly trained to optimize the imaging quality. We should admit that DALL-E and CogView can achieve better details than the method used in our work.
>
> 3),"By the way, what is Mental Search in Fig.1?"-------------------
>  We are sorry for this spelling mistake. In fact, it should be “Petal Search”, a search engine powered by Huawei.
>
> 4),  "The performance of Wukong in Table 3 seems to be out of date. For example, the Flickr30K-CN of Wukong achieves the performance of mean recall(MR) 93.4 other than the reported 75.1. The ImageNet zero-shot top-1 accuracy performance for Wukong is 58% (not 35%) You may update the numbers from Wukong latest arxiv version."------------------------ In Table 3，we have compared with Wukong-ViT-B and Wukong-Swin while our models are ViT-B and RN101, we have not compared with wukong's ViT-Large variant because that's unfair. The figures for Wukong are consistent with its latest version.  And the zero-shot classification performances of Wukong are taken from the first version of Wukong’s paper, because we had not found corresponding models in Version 2 (the latest version 3 was released after our submission). And we found that the methods used in the three versions of Wukong’s paper are a little bit different.  And the version 1 has the most similar methods with our work. So, for fair comparison, we choose to compare with Version 1.

---

> > ### Comment · Reviewer_HiVb · 2022-08-25
> > **Thanks for your response**
> >
> > Thanks for your response. My rate is Good paper, accept

---

### Official Review · Reviewer_6vQb · 2022-07-26
**TaiSu is the largest publicly accessible Chinese image-text pair dataset that can serve as a promising benchmark for VLP tasks.**

**Rating:** 9
**Confidence:** 4

**Strengths:**

1. As the current largest and publicly accessible Chinese image-text pair dataset, TaiSu serves as a promising benchmark for VLP tasks. This will be another useful resource for researchers to use when conducting their own research.

2. The authors provide a framework that automatically filters texts that include sensitive words and words that have no semantic connection with the image, filters image-text pairs with low correlation using cosine similarity, and augments image captioning by combining texts from web and texts generated by machine. TaiSu is the first to augment texts using both sources.

3. The authors perform extensive experiments using the RN101 and ViT-B/32 models pre-trained for CLIP as locked image encoders for a wide variety of VLP tasks. They use more than 10 datasets to do a comparative analysis with TaiSu. The results in the tables are succinctly shown and easy to understand.

4. The paper is organized very well and easy to follow.

**Weaknesses:**

1. In Table 1, the authors list datasets that are primarily in English or Chinese. Are there VLP datasets in other languages? The multilingual LAION-5B which was released recently may be another useful dataset to include in this list.

2. It would be nice to provide an English translation for the Chinese characters in section 3.2 and the 50 most common nouns in Figure 7 of the Appendix.

3. Can the authors please explain why they used RN101 and ViT-B/32 specifically as the models for their experiments?

4. In Figure 2, it is difficult to tell apart between watercolor painting and oil painting. Using a different artistic style can help show clear differences between the four categories.



**Additional Feedback:**

There are several typos that can be fixed:

1. line 6: more than 100M image-text pairs are in English, --> change comma to semicolon (;)
2. lines 8-9: we propose a new framework for automatic dataset acquisition and cleaning and construct a new... --> combine into one sentence
3. line 17-22: Further, we test our dataset on several vision-language downstream tasks. On zero-shot image-text retrieval tasks, TaiSu outperforms Wukong by a considerable margin when the model has a similar scale of parameters. Results demonstrate that TaiSu can serve as a promising benchmark for VLP, both for understanding and generation (generating?) tasks.
5. line 24: foundation model does not make sense
6. line 26-28: Its success is not only attributed to the appearance of advanced models like BERT and ViT, but also to the use of large-scale multi-modal datasets.
7. line 30: in the VLP domain
8. line 31: the English corpus
9. line 34: M6-corpus hasn't been --> has not been
10. line 37: Thus, compared with the English VLP community, there is a lack of large-scale and high-quality public Chinese multi-modal datasets for VLP tasks.
11. lines 39-46: Change Firstly, Secondly, and Thirdly to First, Second, and Third
12. lines 45-46: while it is even mismatched (please fix)

**Clarity:**

The paper is structured very well and easy to follow. Except for a few grammatical errors, the paper is written well. I would suggest to expand all the abbreviated words such as "hasn't" and "it's" to sound more formal.

**Correctness:**

The dataset is constructed in a sound way with filtration and augmentation steps that make sense. The experiments are very thorough and the results support their claim that TaiSu is a promising benchmark for VLP.

**Documentation:**

There is sufficient detail on data collection, creation of TaiSu, and code on github. The supplementary file provides more details about TaiSu, such as basic statistics, most common words, and ethical concerns.

**Ethics:**

The authors address privacy concerns by substituting personal information with special token words and releasing the image URLs instead of the raw images.

**Relation To Prior Work:**

In Section 2, the authors describe related VLP datasets and models and highlight again the dearth of accessible Chinese multimodal datasets.

**Summary And Contributions:**

The authors propose a new framework for automatic dataset acquisition and cleaning to construct TaiSu, a new large-scale and high-quality image-text pair dataset containing 166 million images and 219 million Chinese captions, to address the lack of high-quality public Chinese multi-modal datasets for VLP tasks. In the framework, they perform text-based filtering, cross-model semantic based filtering, and image caption text augmentation to produce a clean and high-quality dataset. They perform extensive experiments using dual-stream VLP models, in particular the RN101 and ViT-B/32, as locked image encoders, and use three types of data to verify the effects of filtering and generated texts. TaiSu shows great performance on a number of vision-language downstream tasks including zero-shot image-text retrieval, zero-shot image classification, and text-to-image generation. The authors plan on making the dataset, pretrained models, and codes public.

---

> ### Author Response · Authors · 2022-08-23
> **The generated samples are updated.**
>
> Thanks for your time spent on the review process. With respect to your review comments, we give the following responses:
>
> 1)	LAION-5B is a huge cross-modal dataset that was recently released, We have added it in our revised manuscript.
>
> 2)	We have added English translations for all the Chinese Words in the revised manuscript. Thanks for your advices.
>
> 3)	The reason for choosing model architectures: ResNet and ViT are two of the most popular visual backbones. There are  many works such as CLIP that were conducted with these two architectures. And  the main reason for choosing  RN101 and ViT-Base is  due to the scale of model’s parameters and computing speed. The number of parameters in a descending order: ViT-L>ViT-B>RN101>RN50.  We choose the two architectures with medium number of paramters. Since we need to train multiple models on different datasets for comparison, it’s costly to train all the possible architectures with limited computation resources.
>
> 4)	As for the generated images,  and we have updated  the  generated samples. Thanks for your advice.
>
> 5)     The grammatical errors are corrected.  Foundation model can be referred to  #[1] Rishi Bommasani, Drew A. Hudson, Ehsan Adeli, Russ Altman, and et al. On the opportunities and risks of foundation models. CoRR, abs/2108.07258, 2021.

---

### Official Review · Reviewer_ae8h · 2022-07-27
**Valuable benchmark**

**Rating:** 7
**Confidence:** 3
**Correctness:** Yes
**Clarity:** Reasonable well

**Strengths:**

The presented Taisu is the largest chinese cross-modal dataset so far and publicly available. It supports many downstream cross-modal tasks. The performance by proposed method is reasonable good compared to SOTA methods.

**Weaknesses:**

1. Product1M in ICCV 2021 and M5product in CVPR 2022 also proposed the Chinese cross-modal dataset. However, the authors miss the comparison with them in Table 1.
2. In sec. 5.4, Taisu presents the Text-to-Image generation results. It misses the comparison with the latest Cogview and Cogview2, which are SOTA image generation method. Could the authors add such comparison?
3. CLIP, BLIP and FILIP are publicly open-sourced. Even though they do not have results on chinese cross-modal dataset in their original paper, the authors are highly encouraged to compare with one of them as they are most widely used.

Product1M: Towards Weakly Supervised Instance-Level Product Retrieval via Cross-modal Pretraining, ICCV 2021.
M5Product: Self-harmonized Contrastive Learning for E-commercial Multi-modal Pretraining, CVPR 2022.

**Additional Feedback:**

N/A

**Documentation:**

Sufficient

**Relation To Prior Work:**

Yes

**Summary And Contributions:**

This paper contributes the largest chinese cross-modal benchmark, named as Taisu. The whole dataset acquisition framework is clearly described and reasonable. The experiments are sufficient.

---

> ### Author Response · Authors · 2022-08-23
> **The comparison with CLIP is added**
>
> Thanks for your constructive comments! According to the issues that you have mentioned, we give the following responses:
>
> 1)	In the Table 1, we list some large-scale cross-modal datasets for the comparison of scale and language, M5product contains 6.3M image-text pairs and Product1M contains about 1M image-text pairs. We have added them in our revised manuscript.
>
> 2)	In our paper, we use our pretrained dual-stream model to guide VQ-GAN to generate images according to the given text prompts. We regard it as a method to visualize the knowledge learned from our proposed dataset. Therefore, we have not compared it with the state-of-the-art text-to-image models. In fact, such a comparison is unfair. The dual-stream models pretrained with contrastive loss focus on the high-level features of images (i.e. semantic) while models like CogView and DALL-E directly learn the distribution of image tokens. So, Cogview and DALL-E can generate images with better details.
>
> 3)	We adopted your advice to compare our pretrained models’ performances on Chinese cross-modal datasets with that of CLIP’s models. Note that CLIP was trained with 400M English text-image pairs and can’t directly process Chinese captions. So, we translate COCO-CN, Flickr8k-CN, Flickr30k-CN and MUGE into English with machine translation. We compare their image-text retrieval performances. The results have been added to the appendix A. Our models out performs CLIP on all the dataset used for testing.These results can reveal the necessity of constructing a large-scale Chinese cross-modal datasets, because the semantic gap between different languages have not been well addressed by machine translation yet.

---

### Official Review · Reviewer_sjpQ · 2022-07-28
**Review to Paper155**

**Rating:** 7
**Confidence:** 4
**Correctness:** It is constructed in a sound way.
**Clarity:** It is fine to me.

**Strengths:**

1. TaiSu is currently the largest Chinese image-text dataset.
2. Compared with previous works (e.g., Wukong), this dataset leverages an image-text retrieval model to clean unpaired data and further improve the correlations between image-text pairs. In addition, some of the images on the dataset have multiple captions generated by an image captioning model, which expands this dataset. This may provide an idea for future work to build and filter a multi-modal dataset.
3. The experiment shows the dataset can help pre-training models can get better performance on many datasets than previous works with the same pre-training method.

**Weaknesses:**

1. Although the number of the images in TaiSu has 50% larger than Wukong, the pre-trained model by TaiSu fails to outperform the one got from Wukong on some downstream datasets.




**Additional Feedback:**

1. Documentation for the dataset should be provided.
2. Details of the paper can be improved. For example, there are some duplicate references, e.g., [9] and [21], [12] and [23].
3. Could you please explain why the pre-trained model from TaiSu fails to outperform the one from Wukong on sone datasets, such as MUGE and Flowers? I am interested in this phenomenon.


**Documentation:**

The intended uses are clear in the main paper. The data collection, URL to the dataset, and licensing are also available, but I fail to find documentation (e.g., datasheets) in the supplementary material.

**Ethics:**

The authors state the potential negative societal impacts.

**Relation To Prior Work:**

The proposed dataset is much larger than previous works, such as Wukong and WudaoMM.
As I state in the Strengths, the authors utilize an image-text retrieval model to remove the image-text pairs which are weakly aligned, and an image captioning model to further augment the dataset.


**Summary And Contributions:**

This paper proposes TaiSu, a large-scale multi-modal dataset that contains over 166M images and 219M Chinese captions, along with a new framework for acquiring and cleaning image-text data. The experiments on downstream tasks (i.e., zero-shot image classification and zero-shot image-text retrieval) show the effectiveness and high quality of TaiSu for vision-language pre-training.

---

> ### Author Response · Authors · 2022-08-23
> **The datasheet is added and the experimental results  are explained**
>
> Thanks for your kindly review comments. The datasheet is indeed missing in the supplementary materials, we will add it in our revised version. In fact, we have detailed the Motivation, Composition, Collection Process, Preprocessing, Cleaning, and Labeling for the construction of TaiSu. We have summarized relevant information and formed a datasheet.  And thank you again for pointing out the mistakes in detail, we will pay attention to them.
> As for the comparison between models pretrained on TaiSu and Wukong datasets, we list the following reasons to explain :
>
> 1)	The number of model parameters is different:  The main goal of the experiments presented in this paper is to validate the effectiveness of the proposed methods for the construction of large-scale cross-modal dataset. And we directly use the hyperparameters used for the training of CLIP’s models. So the hidden state dimension of transformers in our textual encoder is 512 while that is 768 in Wukong’s textual encoder. In another  word, Wukong’s models have much more parameters than ours.
>
> 2)	Image Data augmentation is different: Because of the limitation of compuational resources and the motivation to speed up the training process, the image features used in our paper are pre-extracted without training the parameters of RN101 or ViT-B/32 networks. The models in Wukong are trained with image data augmentation which is helpful for image classification. In other words, we have not applied image data augmentation while Wukong has applied it.
>
> 3)	Distribution of data is different:  It’s easy to understand that the performance on classification task is correlated with the number of occurrences of different categories.  As the data are queried from Web using the most frequent search items, some categories may have more samples than the other. The  query distributions in Taisu and Wukong are very different. What’s more, Wukong and TaiSu are constructed with different processes, their distributions of data may be very different.
>
> 4)	Additionally, MUGE is a dataset used for the retrieval of E-commerce products, where the Chinese captions in it have many differences compared with COCO-CN, Flickr8k-CN, Flickr30k-CN in terms of the structure of sentences and vocabularies.  Our models have better performances on COCO-CN, Flickr8k-CN and Flickr30k-CN, and the sentences in these datasets are similar with the sentences used in our daily life.   As we can see, Swin transformer pretrained on Wukong has achieved the best results on MUGE, while it has the worst performances on the other datasets. It also implies that MUGE has very different characteristics compared with other datasets.
>
> *Note:  Although TaiSu has 166M images, but only 133M images has natural sentences from web, the rest images only have the sentences generated by machine, we can regard them as pseudo labels.

---

> > ### Comment · Reviewer_sjpQ · 2022-08-29
> > **Reponse**
> >
> > Thank you for your response to solve my concerns.
> >
> > May I ask whether the script for downloading the images will be available? It seems that the script currently remains unreleased in the Github repo.

---

### Meta-Review · Area_Chair_2u6s · 2022-09-02

**Recommendation:** Accept
**Confidence:** 5

**Metareview:**

The reviewers are positive regarding the high level of the contribution of the work for the NeurIPS 2022 Track Datasets and Benchmarks. The authors properly addressed all reviewers comments and concerns during the rebuttal period.

---

### Decision · Program_Chairs · 2022-09-16

Accept